# Intervertebral Disc Disease of the Lumbar Spine in Health Personnel with Occupational Exposure to Patient Handling—A Systematic Literature Review and Meta-Analysis

**DOI:** 10.3390/ijerph17134832

**Published:** 2020-07-04

**Authors:** Christofer Schröder, Albert Nienhaus

**Affiliations:** 1Department of Occupational Medicine, Public health and Hazardous Substances, Institution for Statutory Accident Insurance and Prevention in the Health and Welfare Services, 22089 Hamburg, Germany; albert.nienhaus@bgw-online.de; 2Competence Centre for Epidemiology and Health Services Research for Healthcare Professionals (CVcare), University Medical Centre Hamburg-Eppendorf (UKE), 20246 Hamburg, Germany

**Keywords:** Health personnel, occupational exposure, patient handling, intervertebral disc disease, disc degeneration, musculoskeletal disorders

## Abstract

Lifting or carrying loads or working while the trunk is in a bent position are well established risk factors for the development of disc disease of the lumbar spine (LDD). Patient handling is associated with certain hazardous activities, which can result in exposure to heavy loads and high pressure for the discs of the lumbar spine of the nurses performing these tasks. The purpose of this review was to examine the occurrence of work-related LDD among health personnel (HP) with occupational exposure to patient handling activities in comparison to un-exposed workers. A systematic literature search was conducted using the following databases: PubMed, CINAHL, Scopus, and Web of Science. A meta-analysis of odds ratios (OR) was conducted by stratifying for various factors. Five studies reported a higher prevalence for LDD among nurses and geriatric nurses (11.3–96.3%) compared to all controls (3.78–76.47%). Results of the meta-analysis showed a significantly increased OR for LDD among HP compared to all controls (OR 2.45; 95% confidence interval (CI) 1.41, 4.26). In particular, the results of this review suggest that nurses have a higher probability of developing disc herniation than office workers.

## 1. Introduction

Intervertebral disc disease of the lumbar spine (LDD) has been identified as one of the main causes of low back pain that affects almost everyone, especially adults [1]. LDD is widespread and has a multifactorial etiology. In addition, LDD is not easy to diagnose, as radiologic alterations of the disc do not necessarily mean disease. They become a disease when they are associated with typical symptoms. LDD is associated with genetic factors [2], ageing [3,4,5], smoking [2], lack of exercise [6], and obesity [4,7,8]. People develop LDD more frequently while carrying out certain professional activities [9]. Work-related LDD is essentially caused or aggravated by lifting or carrying loads or working with the trunk in a bent position. Special characteristics of lifting or carrying of loads includes load handling operations such as transferring or setting down as well as holding, pulling, or pushing loads. An additional risk is associated with lifting and carrying heavy loads and work in a position that involves extreme bending of the trunk, if they are performed in a twisted posture. An increased risk for the development of LDD has been established for nursing staff and among nursing assistants [10,11,12,13]. Health personnel (HP) with occupational exposure to patient handling activities usually support a considerable part of the patients’ body weight and act in an unfavorable posture, which is characterized by strong bending, lateral inclination, and often simultaneous twisting of the upper body, which leads to a high load on the intervertebral disc [14]. Certain hazardous activities were identified in patient handling where the load is of such an extent that the spinal column of the nursing staff is overloaded [15,16]. Such activities include raising a patient from a lying position to a sitting position in bed or at the bed’s edge; moving a lying patient towards the head of the bed (nurse at the bed’s long side or at the head of the bed) or sidewards in the bed; inclining the head of the bed with a patient lying in it; positioning or removing a bedpan; moving a patient seated at the bed’s edge to a chair; and raising a patient from sitting to standing upright [17,18,19,20]. Moreover, working in an inclined posture, such as washing, mobilizing, or changing beds, is considered to be an additional risk factor for back disorders. Among nursing staff, patient handling is one of the most important risk factors of lumbar spine complaints [21,22,23], but it is also practiced by physiotherapists, rescue service personnel, hospital porters, or outpatient nursing staff, among others.

The terminology of disc disease includes various definitions and pathophysiology. The intervertebral disc connects adjacent vertebrae of the spinal column to each other and absorbs mechanical loads. It enables flexion, extension, bending, and rotation of the spine [24,25]. Intervertebral disc disease leads to instability, stenosis, and deformity of the locomotor segments of the spine [26,27]. Disc degeneration is defined as wear-related, morphological changes in the intervertebral disc tissue. Due to reduced water retention and reduced elasticity, the intervertebral discs are restricted in their ability to absorb and distribute pressure. There is a risk of tearing of the fibrous ring (annulus fibrosus). The consequence is a reduction in the height of the intervertebral space, resulting in a restriction of function and increased stress on the osseous parts of the joint, which leads to arthrosis. In a herniated disc, tissue is displaced or leaks from the nucleus pulposus of the disc through mostly degenerative or rarely traumatic tears in the annulus fibrosus. Disc herniation includes protrusion/bulging of the intervertebral disc in a dorsal direction [28].

In occupational studies low back pain (LBP) is often taken as outcome measure. However LBP can be caused by multiple factors such as stress, sedentary life style, lack of physical fitness, and psychological factors [29,30,31,32,33]. In Germany, LBP is not considered as an occupational disease when LBP is not associated with disc degeneration. Therefore, the outcome for our review was defined as LDD verified by imaging technics or confirmed diagnosis.

To the authors knowledge, there is no systematic review or meta-analysis which surveys the occurrence of disc disease among HP. The purpose of this literature review was to examine prevalence or incidence data of work-related LDD among HP and a comparison group, as reported in individual studies. In addition, the effects of the individual studies were summarized by performing a meta-analysis.

## 2. Materials and Methods

This systematic review was performed in line with the Proposal for Reporting of Meta-analyses of Observational Studies in Epidemiology (MOOSE) [34] (Appendix A).

### 2.1. Search Strategy

A systematic literature search was conducted using the following databases from their beginning up to 2020/01/10. Therefore, no time frame other than the one introduced by the documentation systems was defined for the eligible studies: PubMed; CINAHL (via EBSCOhost); Scopus; and Web of Science. The systematic electronic search did not apply any language restrictions to reduce publication and retrieval bias. Articles published in languages other than English have been translated by native speakers or with internet service providers. The search string was created sensitively by combining the keywords with Boolean operators and adapted respectively to each database with the following keywords: moving lifting patients OR patient handling OR patient transfer OR therapist OR physiotherapist OR health personnel OR nurse OR care worker OR occupation OR allied health OR work related OR working environment OR occupational exposure. This was combined with the linkage AND to the following search string: disc (degeneration OR herniation OR disease OR prolapse OR protrusion OR injury OR displacement OR disorder OR herniated OR bulging OR degenerative) OR spondylosis OR osteochondrosis OR back pain AND (tomography OR magnetic resonance imaging OR MRI OR CT). A detailed description of the general search strategy is provided in Appendix A. Search results were exported to EndNote X9 (software companies, city, state abbr. if USA/Canada, country) [35] to conduct the selection process. A registration of the full search strategy including all keywords at PROSPERO was not possible as PROSPERO did not accept registrations for scoping reviews, literature reviews, or mapping reviews at the time of writing this review. Systematic reviews were screened for further possible hits. A manual search of the reference lists of all included studies after title and abstract screening and of the reference lists of related key articles was performed to supplement the electronic search. We contacted three authors [36,37,38] to obtain further data for the meta-analysis of which Michaelis et al. provided us the required data. Studies using the outcome low back pain in combination with orthopedic radiological findings or data including confirmed diagnosis were included according to the population, intervention, comparison, outcome, study-design (PICOS) strategy, where “I” was replaced for “E” (exposure) [39].

P: Health personnelI (E): Manual patient transfer, patient transfer with small or technical patient handling aids, performing nursing activities, job-title based exposureC: general population, other occupational groups, subgroups within HP, or self-comparison over timeO: Specific disease of the lumbar spine such as disc degeneration, disc herniation, disc protrusion, disc bulging, spondylosis, modic changes, and endplate changesS: Any design involving a comparison group

### 2.2. Study Selection

The study selection by title/abstract and full text was carried out independently by two reviewers (C.S. and A.N.) with regard to the research question defined a priori, in line with the defined inclusion and exclusion criteria. Disagreements were resolved by discussion in consensus conferences. In case of nonconformity a third assessor was consulted.

### 2.3. Quality Assessment

The Newcastle–Ottawa Scale (NOS) was used for assessing the quality of non-randomized studies in meta-analyses [40]. For performing a quality assessment of cross-sectional studies, the NOS for cohort studies has been adapted by Modesti et al. [40], which is the version we edited for this review. After evaluating the selection, comparability, and outcome/exposure categories, the quality was assessed using the thresholds for converting the NOS into the Agency for Healthcare Research and Quality (AHRQ) standards “good” (+++), “fair” (++) and “poor” (+). In addition, a risk of bias assessment was performed using RevMan 5.3 software [41].

### 2.4. Data Extraction

Data of the included studies was extracted by one author and checked for accuracy by a second author. The data extraction took into account, using a standardized data extraction form: study design; aim of the study; year(s) and country of data collection; sample size in analyses; study population including occupation, sex, and age in years of the exposure and comparison group; explanatory variables; method of outcome assessment; prevalence or incidence of degenerative findings; and effect measures. Where percentages rather than absolute numbers were given in the primary study, the absolute numbers for cases and controls were derived from them or vice versa. Where no effect measures had been calculated by the authors of the original study, but sufficient data was available, we calculated (prevalence) odds ratios using Review Manager Version 5.3 software [41]. Where data was missing or incomplete for extraction and analysis, authors were contacted, and the additional information was requested. Where the required data was not obtainable, the study was excluded from the meta-analysis.

### 2.5. Data Synthesis and Statistical Analysis

RevMan 5.3 software provided by the Cochrane Collaboration was used for data analysis [41]. Odds ratios (OR) were calculated using prevalence rates from the original manuscripts. Relative risks (RR) were converted into OR with 95% confidence interval (95% CI). Heterogeneity was tested using chi-square tests (*p* < 0.1 means heterogeneity). The I^2^ statistic was also used to evaluate the heterogeneity (0% to 40% might not be important; 30% to 60% may represent moderate heterogeneity; 50% to 90% may represent substantial heterogeneity; 75% to 100% considerable heterogeneity) [42]. The pooled effect measure was estimated using random effects models, because we assumed that a distribution of effects depends on the study characteristics [43]. *p* < 0.05 was considered to be statistically significant. Further stratification was performed relative to types of participants/comparators and study quality. A sensitivity analysis was performed by investigating the stability of the pooled estimate with respect to each study by excluding single studies from the meta-analysis.

### 2.6. Publication Bias

Publication bias was judged with a funnel plot. In addition, the possibility of publication bias was tested using Egger’s linear regression in SPSS version 25 [44]. The level of significance for asymmetry was based on *p* < 0.1. The calculated intercept was given with a 90% confidence range.

### 2.7. Ethical Consideration

There was no contact with any study participants at any time, therefore no ethics committee was required for a systematic review of published literature.

## 3. Results

After removing duplicates, the database search resulted in 3053 articles. Of these, 90 were selected as potentially eligible based on their title and abstract. After full text screening, seven articles met the inclusion criteria and were included in the systematic review [36,37,45,46,47,48,49]. Of these, five studies could be included in the meta-analysis [37,45,46,47,48]. The flow chart of the study selection process is summarized in Figure 1.

### 3.1. Study Quality

The quality assessment demonstrated an overall fair level of quality (Table 1). Detailed results of the quality assessment with NOS including converting into AHRQ-standards are given in Appendix A. Two studies were assessed as good [36,37], two as fair [45,49], and three studies were of poor quality [46,47,48]. Poor quality studies were less satisfactory in the category comparability, which had a noteworthy effect on the overall study quality. The seven included studies were better rated in the categories selection and outcome. Nevertheless, the risk of a selection and a performance bias was rated as high in all included studies, due to the fact that no randomization based on the research question was possible (Table 2). The allocation was based on the job title and could therefore not be concealed. The risk of detection bias could not be assessed in any of the studies because no sufficient information on the outcome assessment was reported. In two studies [47,49], an attrition bias was found to be high due to losses in follow-up surveys. A reporting bias was rated as low in all studies. A detailed description of the risk of bias assessment is provided in Appendix A.

### 3.2. Study Characteristics

All included studies were observational studies, three cross-sectional studies [46,47,48], one case control study [37], and three cohort studies [36,45,49] which analyzed prevalence or incidence rates and/or occupational risk factors of LDD among HP (Table 3).

Six of seven studies were published in English and one in German between 1987 and 2017; three of these, however, had been published in the last decade. The countries of data collection were Finland (*n* = 1), Germany (*n* = 2), Italy (*n* = 1), Japan (*n* = 1), Taiwan (*n* = 1), and United Kingdom (*n* = 1). The studies were based on similar research objectives. In four of seven studies, different risk groups were examined for disc disease; in two studies, various musculoskeletal disorders were investigated in the study population; in one study, the potential role of MRI in the evaluation of LBP was examined.

The analyzed sample size ranged from 73 to 15,658. In six studies, nurses represented the exposure group, but in two of these studies the exposure groups had been supplemented by related medical workers [45] or geriatric nurses (40%) [37]. Savage et al. examined hospital porters and ambulance men as exposure group.

The comparison groups of three included studies were represented by personnel with sedentary jobs (white-collar workers; university staff (computer users); office staff), which was classified as office workers [45,47,48]. In another three studies the exposure groups were compared to the general population [36,37,46]. These were, for one thing, employed persons who were not exposed to spine-burdening occupational activities at all [46] and, for another thing, employed and non-employed persons who have never been exposed to patient handling [36,37]. The prospective study by Makino et al. did not have a separate comparison group, but the authors assessed the development and process of LDD in nurses within the group [49]. The authors examined the prevalence of LDD in nurses at baseline and again after an average of ten years. Furthermore, they compared the activity of nurses in different wards and the number of years of service as potential risk factors for LDD.

The subjects of all studies were generally young in age. The youngest and oldest mean age was 20.9 (range 20–22) [49] and 47.9 (SD 11.5) years [37]. Three studies indicated the age range of their study populations (20 to 59) [45,46,47]. In the study by D’Agostin and Negro [48], the mean age of the exposure group was slightly higher than of the comparison group (42.3, SD 10.2 vs. 38.7 SD 12), while in the study by Chung et al., the mean age of the exposure group was slightly lower than of the comparison group (33.98, SD 7.68 vs. 34.01, SD 7.69). Matching by age and gender was performed in three studies [36,37,45], with Makino et al. including only young women with an average age of 20.9 (20–22) years anyway. The participants of the exposure group were predominantly female (76.8–98.65%). While Heliovaara and Makino et al. only included females in their study population, Savage et al. included only males. The female percentage in the study by D’Agostin and Negro was slightly higher in the exposure group than in the comparison group (76.8% vs. 60.0%) [48].

Medical imaging such as MRI was performed in five studies [37,46,47,48,49], whereas one study [48] also analyzed computer tomography, ultrasound, electro-neurographic, and, X-ray images to assess degenerative changes [46,47,49], disc herniation and protrusion [37], or musculoskeletal disease [48]. Two studies used the International Classification of Diseases (ICD)-8-codes 725.10 or 725.19 (herniated lumbar disc) and ICD-9-codes 721.3 (herniated intervertebral disc) and 722.10 (lumbar spondylosis) [36,45].

### 3.3. Prevalence/Incidence Rates and Effects Measures of LDD in Individual Studies

All studies reported a higher prevalence of LDD for HP with occupational exposure to patient handling compared to controls, with the exception being hospital porters, who had a marginally lower prevalence of LDD than office workers (43.75% vs. 43.86%; Table 3). The highest reported prevalence of LDD for nurses was 96.3% [46] and the lowest 11.3% [48]. Heliovaara reported a prevalence of 29.03% of LDD among nurses and related medical workers [45]. The prevalence of LDD was higher among nurses and geriatric nurses with more than ten years of service than those with fewer years of service (76.47% vs. 62.5%). Makino et al. reported a prevalence of 31% of LDD among nurses at the average age of 20.9 years at baseline [49]. The prevalence of LDD was 43.75% among hospital porters and 50% among ambulance men [47]. Chung et al. indicated an incidence of 1.45 for herniated intervertebral discs among nurses compared to 0.64 in the general population and an incidence of 1.07 for lumbar spondylosis among nurses compared to 0.81 in the general population. The prevalence of LDD among office workers in two different studies was 49.18% [37] and 76.47% [46], while the prevalence of LDD among the general population ranged from 49.18% [37] to 76.47% [46].

All included studies that examined nurses showed an increased OR or RR for LDD among nurses compared to the reference groups [36,37,45,46,48]. Nurses in intensive care units (ICU) and operating theatres (OP) had a non-significant lower risk of disc degeneration than clinic nurses (RR = 0.79; 95% CI 0.46, 1.37) [49]. Among nurses, the OR of 8.00 (95% CI 1.69, 37.82) was highest in Hartwig et al. compared to the general population [46]. Chung et al. reported an increased OR of lumbar spondylosis (OR 1.36; 95% CI 1.00, 1.84) and disc herniation (OR 2.48; 95% CI 1.82, 3.38) among nurses compared to working (non-nursing) and non-working controls [36]. The OR in the study by D’Agostin and Negro was statistically significant increased among nurses OR 3.11; 95% CI 1.28, 7.56) [48] compared to university staff (computer users), and the authors of the longitudinal study by Heliovaara indicated a statistically non-significant increased risk among nurses for disc herniation compared to white-collar workers (RR 2.2; *p* ≥ 0.05). Michaelis et al. reported a significantly increased OR where nurses and geriatric nurses had been employed for more than ten years (OR 3.3; 95% CI 1.1, 10.4), but the results showed an insignificantly smaller OR when nurses and geriatric nurses had worked for less than or equal to ten years (OR 1.72; 95% CI 0.40, 7.27) [37]. Makino et al. indicated a non-significant increased risk in nurses who had been working for more than five years (RR 1.30, 95% CI 0.73, 2.30) [38]. Hospital porters did not show a higher or lower likelihood compared to office staff (OR 1.00; 95% CI 0.33, 3.04), whereas ambulance men showed a slightly non-significant increased OR of 1.28 (95% CI 0.49, 3.34) compared to office staff [47].

### 3.4. Meta-Analysis

The main results are shown as plots in Figure 2a. A total of five studies were included in the pooled analysis. The two subgroups (hospital porters and ambulance men) in Savage et al. [47] were combined into one group. In this meta-analysis, the exposure group of the study by Michaelis et al. [37] was compared to office workers instead of several occupational groups as in the original study with the aim of keeping the control group in the meta-analysis as homogeneous as possible.

The pooled analysis of all studies showed a significantly increased OR of 2.45 (95% CI 1.41, 4.26) for LDD among HP compared to all controls, with moderate evidence of heterogeneity (χ2 = 6.60, *p* = 0.16, I^2^ = 39%).

When excluding hospital porters and ambulance men, the OR increased to 3.02 (95% CI 1.84, 4.95) among nurses and geriatric nurses compared to all controls, without significant evidence of heterogeneity (χ2 = 2.70, *p* = 0.44, I^2^ = 0%; Figure 2b), whereas hospital porters and ambulance men showed a non-significant OR of 1.16 (95% CI 0.51, 2.61) compared to office workers (Figure 2a). A further stratification through the sole inclusion of office workers as a comparison group led to a stratification of the dependent variable to disc herniation. Here, nurses and geriatric nurses showed a significantly increased OR of disc herniation compared to office workers (OR 2.70; 95% CI 1.61, 4.55) Figure 2c).

Including two studies of good and fair quality, the OR of 2.46 (95% CI 1.29, 4.68) was significantly increased among nurses and geriatric nurses compared to office workers (Figure 2d). Studies of poor quality showed a non-significantly increased OR of 2.68 (95% CI 0.97, 7.38) among HP compared to controls, with significant heterogeneity (χ2 = 5.78, *p* = 0.16, I^2^ =65%; data not shown).

### 3.5. Heterogeneity and Sensitivity Analysis

Moderate heterogeneity was present when all studies were pooled, but there was no heterogeneity in any of the subgroups (Figure 2a–d). Furthermore, individual studies were subsequently excluded one after the other from the analysis to investigate their influence on the pooled estimate (Appendix A).

### 3.6. Publication Bias

The funnel plot did not show evidence of publication bias (Appendix A); however, the number of publications presented is low. Egger’s linear regression did not show significant evidence of funnel plot asymmetry (intercept 0.322; 90% CI −1.870, 2.531; *p* = 0.770).

## 4. Discussion

This systematic review addressed the literature on the association between health personnel with occupational exposure to patient handling and the presence of intervertebral disc disease of the lumbar spine. All seven analyzed studies were observational studies. Nurses were predominantly investigated as the exposure group, whereas office workers or the general population were investigated as the comparison group. The studies had different objectives, but six of seven studies allowed a comparison of occupational groups in cross-section or follow-up, whereas one study compared subgroups within HP and compared them over time [49]. The meta-analysis showed that HP have a significantly increased occurrence of disc disease; however, it must be noted that there was moderate heterogeneity when all studies were pooled. The subgroup analysis, in particular, showed an increased occurrence of disc herniation in nurses compared to office workers. Nevertheless, the results should be interpreted with caution. It may be difficult to compare the results of one hospital with other hospital types. There might be differences in the working methods of different countries, or working structures may have changed over the decades, so that, for instance, nurses 30 years ago were much more exposed than they are nowadays.

The quality of the included studies in the meta-analysis was rather poor, since two studies of good and fair quality had to be excluded. On the other hand, two studies of good and fair quality showed the same increased occurrence of disc herniation in nurses and geriatric nurses.

The quality was assessed as good and fair, respectively, in two studies each, and as poor in three studies. The inferior quality of the studies was due to a lack of comparability of the exposure and comparison group, because two studies did not describe how the comparison group was selected, while in another study, the loss of follow-up was high. The age of the participants was either often not taken into account when comparing the groups or it was not reported separately with respect to exposure and comparison group. Since age and other risk factors are relevant for the development of disc disease, confounding could not be eliminated completely. Prevalence varies considerably between the studies. Hartwig et al. [46] reported the highest prevalence among nurses. However, a selection bias may exist in this study, since the data was derived from assessments within the framework of a procedure for the identification of occupational diseases. As a result, cases with degeneration of several intervertebral discs have better chances of recognition and are therefore likely to be reported more often. A high prevalence also existed in the comparison group, as these were patients with existing back problems. In addition, there was no indication of how the controls were selected. For these reasons, the study quality was rated as poor. Compared to office workers, ambulance men or hospital porters had no increased prevalence of MRI changes. This may be due to the fact that although the activities of this occupational group include moving patients between different areas of the hospital, lifting or carrying loads or working with the trunk in a bent position are not common. The quality of this study was poor because the sample size was small, and recruitment was not described.

Two included studies examined LDD in nurses according to years of employment. The OR of LDD in nurses and geriatric nurses almost doubled after ten years of service [37]. In another study, the risk for LDD was only slightly increased among nurses after five years of service compared with nurses with less than or equal to five years employment [49]. In this study population, 31% already had LDD at the average age of 20.9 years at baseline. Nevertheless, in these subjects LDD progressed rapidly compared to those who did not have LDD at baseline of the study. The same studies classified the exposure group according to ward or described where they had worked. One reported a non-significantly lower risk for LDD in nurses working in the OP or ICU compared to clinic nurses [49]. This may be due to fewer transfers in the ICU, which are a burden on the lumbar spine. OP nurses may have fewer patients to handle, or more personnel may be available for patient transfers. In another study, the population included 40% geriatric nurses in addition to nurses from the department of internal medicine, surgery, orthopedics, pediatrics, surgery, and diagnostics; but the case numbers were too small to allow a differentiated evaluation [37].

A gender-specific evaluation was not possible because the included studies did not provide relevant data for stratification. However, there were studies included that (almost) solely included female nurses. An incidence study reported a significantly increased likelihood of disc herniation and a slightly increased likelihood of lumbar spondylosis among female nurses compared to the general population [36]. Another incidence study which could not be included in this review examined occupations stratified by gender with a high risk of developing an occupational disease that included disc-related diseases of the lumbar spine as a result of many years of lifting and carrying heavy loads or working in positions that involves extreme bending of the trunk [50]. The results of this study showed that health personnel who perform transfers and work in an extreme bent posture are more likely to suffer from disc disease than other occupational groups. The gender-specific analysis showed that for these occupational groups, the annual incidence rate per 100,000 employees is considerably higher in women than in men [36].

An age-specific analysis could not be performed either, as the included studies did not provide data needed for stratification. Therefore, no statement about the age of the employees and the occurrence of disc disease can be made based on these studies. Makino et al., however, included only young women in the study population, 31% of whom already suffered from LDD at an average age of 20.9 years at the beginning of the study, which is consistent with other studies. Cheung et al. reported that 40% of their study population aged under 30 years had LDD on MRI, but the proportion of people with LDD increased with age [4]. Savage et al. divided their study population into two age groups, but the numbers for HP and office workers were too small for calculations. In the entire study population of Savage et al., which additionally included other occupational groups (car production workers and draymen), disc degeneration was most frequent in L5/S1 and significantly more frequent (*p* < 0.01) in the older age group (31–58 years; 52%) than in the younger age group (20–30 years; 27%). There is ample evidence that LDD is more prevalent in the older population [51,52]. Compared to adults of working age, older adults are more likely to develop certain LBP pathologies, such as osteoporotic vertebral body fractures, tumors, spinal infections, and lumbar spinal stenosis. [53]. An earlier onset of illness in exposed workers is also an important observation that justifies preventive measures.

Among those studies that did not report required data for meta-analysis, one allowed the prevalence to be calculated from the available data. Unfortunately, this process was not feasible for the study by Chung et al. [36], and the authors did not respond to the request for the required data. This study could therefore not be included in the meta-analysis. The authors of the study by Schenk et al. [38] did not respond either. Therefore, the study could not be included in the meta-analysis.

This review has some limitations which need to be addressed. There were only a small number of five studies included in the meta-analysis. A test of publication bias was performed, although we recognized that this may not be accurate for small sample sizes. Although linear regression according to Egger et al. gave no significant indication of a publication bias, the ability to interpret is limited. Another limitation of the included studies is the confounder control. Nurses may spend much of their working time standing and walking around, which in itself means stress. Also, both exposed and non-exposed workers through patient handling may have been engaged in outside work activities. Staff with sedentary jobs may have engaged in other physical activities after work. We assume that LBP and LDD are caused by multiple factors. This renders the analysis of risk factors difficult and non-differential misclassification is a likely bias. In order to eliminate these confounding factors, both occupational and non-occupational spine straining activities would have to be considered for both groups.

The number of studies was too small for further subgroup analysis. More studies, especially studies of higher quality, are needed to prove potential differences bewteen subgroups. Occupational disc disease caused by patient handling affects different occupational groups in health care, but the exposure groups of the included studies were almost all nurses. We would have liked to include several occupational groups in both, the qualitative and quantitative analysis. For example, Wang and colleagues reported an increased risk of disc herniation and spondylosis in both professional nurses and physiotherapists compared to dentists [41].

HP, which is too often engaged in moving or carrying patients may have associated LDD, while nurses who are only occasionally or rarely engaged in patient handling may not be associated with LDD. HP should take technical or organizational preventive measures. Small or technical patient handling aids might relieve the lower back, however using these methods require considerably more time for nursing care than currently scheduled; especially as the preparation and postprocessing when using technical aids seems to be very time-consuming, in contrast to the actual implementation [54]. Another approach is to avoid or minimize these activities if possible, which could require the deployment of more staff.

A limitation of all included studies is the lack of accurate exposure measurement and outcome definition. All exposure groups were assumed to be exposed to patient handling due to their occupation. The number of lifting and carrying tasks is not negligible in the exposure assessment per working shift over time. We did not know whether the study participants used small or technical patient handling aids. On the other hand, there seems to be no convincing evidence for the preventive benefit of using small aids to reduce musculoskeletal complaints and diseases in patient handling persons [55,56]. A systematic review reported on the difficulty of defining the term disc disease, disc degeneration etc. in studies. An imprecise definition of disease patterns might lead to inconsistencies and challenges for a clear and precise communication in medicine and science [57].

The results of two studies [36,45] using claim files are difficult to generalize, as not all nurses suffering from MSDs seek medical help and would therefore be listed in the claim files. Radiological changes are not always clinically relevant. Degenerative findings in MRI are not necessarily associated with the presence or severity of LBP. The size and type of herniated disc and the location and presence of nerve root compression, which were important in terms of morphological changes, are not related to the patient’s outcome [58]. Most patients with acute LBP, with or without radiculopathy, show significant improvements in pain and function in the first four weeks. Most cases of radiculopathy are self-limiting, and symptoms subside over weeks to months. Most cases of herniated discs resorb or recede eight weeks after symptoms begin [59]. Radiologic alterations are not always clinically relevant. However typical clinical symptoms and disc alterations are associated. Therefore, it is noteworthy to study whether alterations of the discs occur more frequently or earlier in certain working groups.

A high risk of a selection bias or no reporting on how participants were recruited existed in all studies. In numerous studies, the sample size, especially in the exposure group, was small.

This systematic review also has some strengths. To our knowledge, this is the first systematic review which surveys the occurrence of disc disease in HP, based on conducting a meta-analysis of pooled odds ratios. The broad spectrum of literature research, such as no language or time restrictions, is a strength of this review. We therefore did not expect any significant impact due to potentially undiscovered studies. We could include five studies that performed MRIs for outcome assessment, which we found to be a more valid method than disease classification following ICD.

## 5. Conclusions

This review presents the existing studies on the prevalence or incidence of LDD among HP compared to controls. The results of this review suggest an association between HP with occupational exposure to patient handling and LDD. The meta-analysis showed a statistically significant increase in occurrence of disc herniation among nurses and geriatric nurses compared to office workers. In contrast, ambulance men and hospital porters did not show a higher prevalence of disc disease compared to office workers. Disc disease occurs more frequently in nurses who have been in this employment for more than ten years. Exposed HP should take preventive measures to reduce the risk of developing LDD. Of the rare existing studies, only a few were of good quality. More high-quality research is needed in order to describe the occupational risk of LDD in HP. For example, prospective studies with a sufficient number of participants and an adequate exposure and confounder assessment are required.

## Figures and Tables

**Figure 1 ijerph-17-04832-f001:**
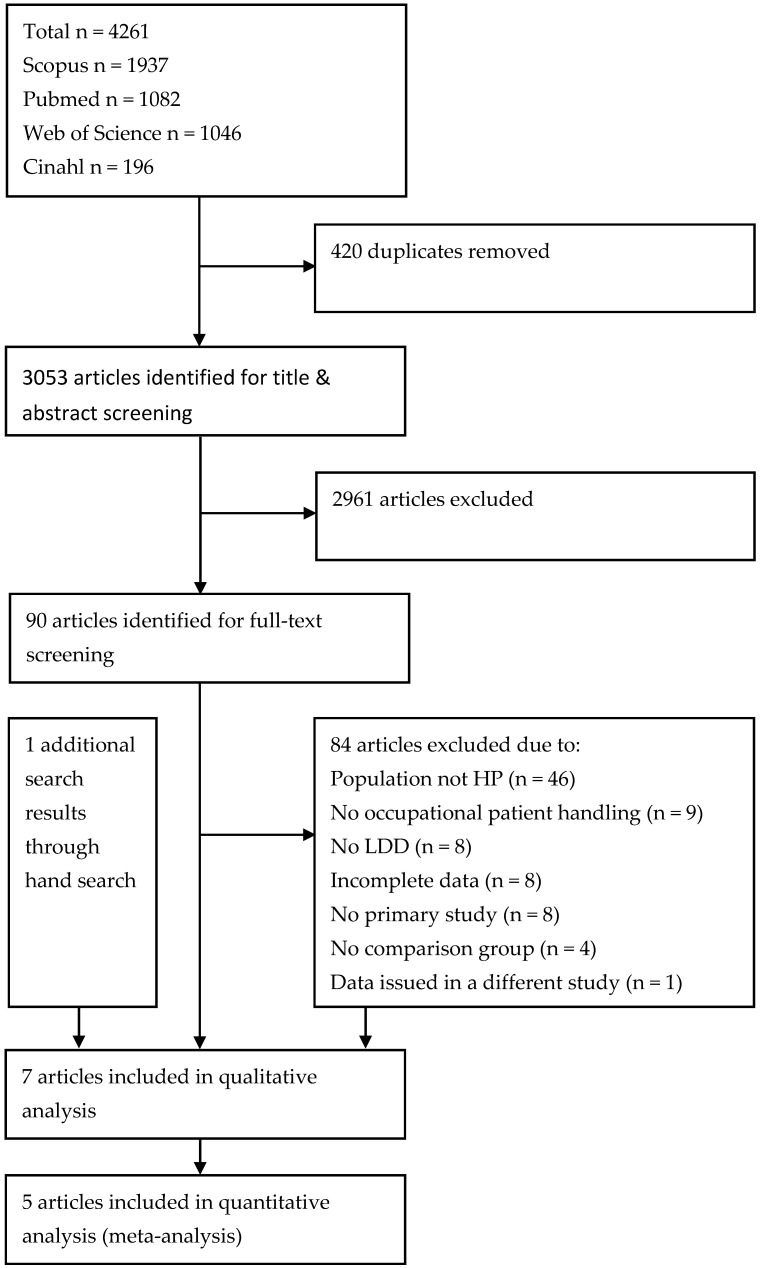
Flow diagram of study selection. HP, Health personnel; LDD, disc disease of the lumbar spine.

**Figure 2 ijerph-17-04832-f002:**
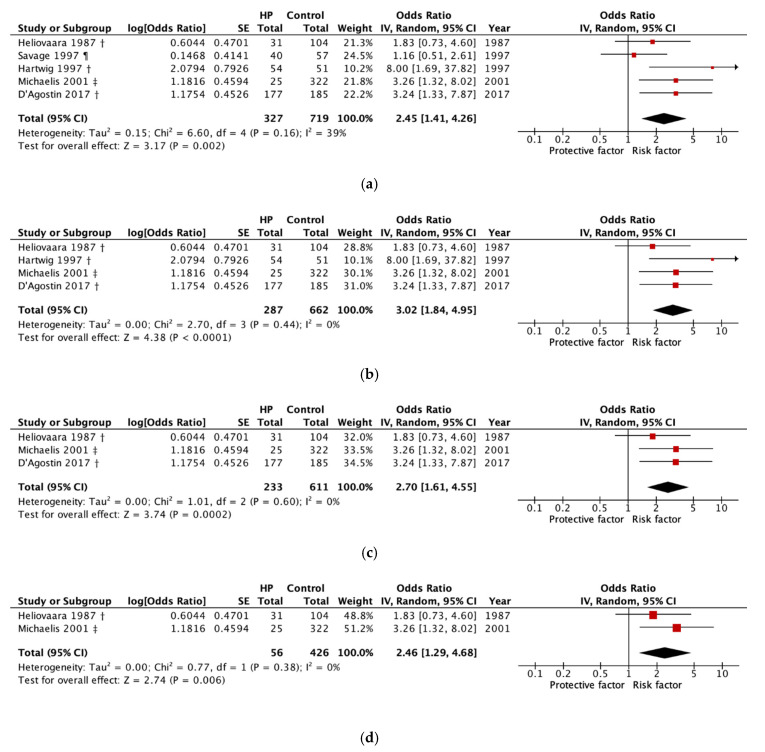
Forest plot of studies included in the quantitative synthesis in disc disease of the lumbar spine among (**a**) health personnel with occupational exposure to patient handling compared to all controls (dependent variable: LDD) (**b**) nurses and geriatric nurses compared to all controls (dependent variable: LDD); (**c**) nurses and geriatric nurses compared to office workers (dependent variable: disc herniation) (**d**) high and fair quality studies (dependent variable: disc herniation). † Nurses; ‡ Nurses & geriatric nurses; ¶ Hospital porters and ambulance men. CI, confidence interval; df, degrees of freedom; HP, health personnel; IV, inverse variance; LDD, disc disease of the lumbar spine; SE, standard deviation.

**Table 1 ijerph-17-04832-t001:** Quality assessment of the included studies.

Author, Year	Selection (Max. 4 +)	Comparability (Max. 2 +)	Outcome (Max. 3 +)	Study Quality *
Heliovaara, 1987 [45]	+++	++	++	++
Hartwig et al., 1997 [46]	-	-	+	+
Savage et al., 1997 [47]	++	-	+++	+
Michaelis et al., 2002 [37]	+++	++	++	+++
Chung et al., 2013 [36]	+++	++	+++	+++
D’Agostin & Negro, 2017 [48]	++++	-	+++	+
Makino et al., 2017 [49]	++	+	+++	++

* AHRQ standard: good (+++), fair (++), and poor (+).

**Table 2 ijerph-17-04832-t002:** Risk of bias assessment of the included studies.

Author, Year	Random Sequence Generation (Selection Bias)	Allocation Concealment (Selection Bias)	Blinding of Participants and Personnel (Performance Bias)	Blinding of Outcome Assessment (Detection Bias)	Incomplete Outcome Data (Attrition Bias)	Selective Reporting (Reporting Bias)
Heliovaara, 1987 [45]	High	High	High	Unclear	Low	Low
Hartwig et al., 1997 [46]	High	High	High	Unclear	Low	Low
Savage et al., 1997 [47]	High	High	High	Unclear	High	Low
Michaelis et al., 2002 [37]	High	High	High	Unclear	Low	Low
Chung et al., 2013 [36]	High	High	High	Unclear	Low	Low
D’Agostin & Negro, 2017 [48]	High	High	High	Unclear	Low	Low
Makino et al., 2017 [49]	High	High	High	Unclear	High	Low

**Table 3 ijerph-17-04832-t003:** Summary of articles included in the systematic review

				Exposure Group		Comparison Group					
Author (Year), Study Design	Aim of Study	Year and Country of Data Collection	Sample Size in Analysis *N* (F (%))	Occupation *N* (F (%))	Age in Years (Range or Mean (SD) At Baseline)	Occupation *N* (F (%))	Age in Years (Range or Mean (SD) at Baseline)	Explanatory Variable	Outcome Assessment	Prevalence of Degenerative Findings *N* (%)	Effect Measure (95% CI) or (*p*-Value)
Heliovaara (1987), Longitudinal study [45]	Identify risk groups for herniated lumbar intervertebral disc or sciatica and to generate causal hypotheses	1966–1972, (follow-up end 1980), Finland	135 (100)	Nurses and related medical workers 31 (100)	20–59 ^#^	White-collar workers 104 (100)	20–59 ^#^	Job-title (NYK 1983)	ICD-8 725.10 or 725.19	*Herniated lumbar disc*: Nurses and related medical workers 9/31 (29.03); White-collar workers 19/104 (18,27)	RR ^a^ = 2.2 (*p* ≥ 0.05)
Hartwig et al. (1997), cross-sectional study [46]	Clarify if different degeneration patterns occur in burdened and non-burdened patients	1994–1995, Germany	105 (*n*/a)	Nurses 54 (*n*/a)	35–50 ^#^	General population, (no spine-burdening activity) with chronic back problems 51 (*n*/a)	35–50 ^#^	Job-title	MRI	*Disc degeneration:* Nurses 52/54 (96.30); General population with chronic back problems 39/51 (76.47)	OR ^b^ = 8.00 (1.69, 37.82) * (RevMan)
Savage et al. (1997), cross-sectional study [47]	Undertake a critical review of the potential role of MRI in the evaluation of LBP	≤1997, United Kingdom	73 (0.0)	Hospital porters 16 (0.0)	20–58 ^#^	Office staff 57 (0.0)	20–58 ^#^	Job-title	MRI	*Disc degeneration, disc herniation, facet joint hypertrophy or evidence of nerve root compression*: Hospital porters 7/16 (43.75); Ambulance men 12/24 (50.00); Office staff 25/57 (43.86)	OR ^b^ = 1.00 (0.33, 3.04) (RevMan)
			81 (*n*/a)	Ambulance men 24 (0.0)	20–58 ^#^	Office staff 57 (0.0)	20–58 ^#^				OR ^b^ = 1.28 (0.49, 3.34) (RevMan)
Michaelis et al. (2001), case control study [37]	Identify overrepresented occupational groups in patients with detectable damage to the intervertebral discs.	1990–1992, Germany	677 (*n*/a)	Nurses and geriatric nurses 8 (*n*/a)	47.9 ^$^ (11.5)	Working (not nursing) general population ^¥^ 669 (*n*/a)	47.9 ^$^ (11.5)	Years of Service <10	MRI/CT	*Herniated disc*: Nurses and geriatric nurses <10: 5/8 (62.5); ≥10: 13/17 (76.47); Working general population ^¥^ 329/669 (49.18)	OR ^b^ = 1.72 (0.40, 7.27)
			686 (*n*/a)	Nurses and geriatric nurses 17 (*n*/a)				Years of Service ≥10			OR ^b^ = 3.36 (1.08, 10.41)
D’Agostin & Negro (2017), Cross-sectional study [48]	Gain insight into the prevalence of MSDs in nursing	2011–2012, Italy	362 (68.23) ^$^	Nurses 177 (76.8)	42.3 (10.2)	University staff (computer users) 185 (60.0)	38.7 (12.0)	Job-title	MRI, CT, US, ENG, X-ray	*Lumbar disc herniation*: Nurses 20/177 (11.30); University staff 7/185 (3.78)	OR ^b^ = 3.11 (1.28, 7.56) * (RevMan)
Chung et al. (2013), prospective incidence study [36]	Assess the incidence of MSDs among a Taiwanese nurse cohort compared with non-nurses	2004–2010, Taiwan	15,658 (98.65) ^$^	Nurses 3914 (98.65) ^$^	33.98 (7.68) ^$^	Working (not nursing) and non-working subjects 11,744 (98.65) ^$^	34.01 (7.69) ^$^	Job-title (NHIRD)	ICD-9-CM (721.3) ICD-9-CM (722.10)	*Herniated intervertebral disc*: Nurses: 1.45; Working (not nursing) and non-working subjects: 0.64	OR ^b^ = 2.48 (1.82, 3.38) *
										*Lumbar spondylosis*: Nurses: 1.07; Working (not nursing) and non-working subjects: 0.81	OR ^b^ = 1.36 (1.00, 1.84) *
Makino et al. (2017), prospective cohort study [49]	Clarify the process and features of lumbar disc degeneration progression in young women	1996–2003 (9,8 (7–14) follow-up), Japan	345 (100) at **baseline**	Nurses 345 (100)	20.9 (20–22)	--	--	Job-title	MRI	*Disc degeneration*: 107/345 (31.0)	--
			84 (100) at **follow-up**	Nurses (OP/IC) 51 (100)	30.6 ^#^ (28–35)	Nurses (Clinic or others) 33 (100)	30.6 ^#^ (28–35)	OP or IC ward		Disc degeneration: (n/a)	RR ^b^ = 0.79 (0.46, 1.37)
				Nurses *n*/a (100)		Nurses *n*/a (100)		Years of service >5		Disc degeneration: (*n*/a)	RR ^b^ = 1.30 (0.73, 2.30)

Abbreviations: 95% CI, 95% confidence interval; CT, Computed Tomography; ENG, Electro-Neurographic; F, female; ICD, International Classification of Diseases; IC, Intensive care; MRI, Magnetic Resonance Imaging; MSD, musculoskeletal disorder; n/a, not available; NHIRD, The Taiwan National Health Insurance Research Database; NYK, Nordic Standard Classification of Occupations; OP, Operation room; OR, Odds ratio; RR, Relative risk; SD, Standard deviation; US, Ultrasound; ^a^ Multivariate analysis: occupational activity, self-reported work incapability, occupational category, smoking, chronic cough, symptoms suggesting psychic distress, and use of analgesics; ^b^ Univariate analysis; ^$^ own calculation; ^#^ Not differentiated between test and control group; * *p* < 0.05. ^¥^ Storemen (*n* = 42); metalworking industry (*n* = 117); building trade (*n* = 48); cleaning personnel (*n* = 14); motorists, transport machine operators (*n* = 41); trade (not metalworking; *n* = 71); sitting and forced work posture (*n* = 46); educators (*n* = 12); agriculture and forestry (*n* = 26); hotel, catering, and retail trade (*n* = 103); technical professions (*n* = 25); administrations and office workers (*n* = 322); health professions (not nursing; *n* = 27); canal workers (*n* = 2); car driver (not heavy haulage; *n* = 14). RevMan, Effect measures were not given by the original study and have been calculated by Review Manager 5.3.

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
