# Peer review of "Intervertebral Disc Disease of the Lumbar Spine in Health Personnel with Occupational Exposure to Patient Handling—A Systematic Literature Review and Meta-Analysis"

_ijerph, 2020, doi:10.3390/ijerph17134832_

Round 1
Reviewer 1 Report
The authors present an exciting research study to answer a question that has been extensively addressed in occupational health services around the world for decades: is load management associated with lumbar disc disease in healthcare personnel? To do this, the authors have carried out a systematic review of the literature complemented by a meta-analysis.
The systematic review is correctly planned and adequately carried out, as well as the meta-analysis is well addressed, and its results well presented. The writing of the manuscript is clear and intelligible.
However, there are a number of comments I would like to make below:
For the evaluation of the quality of the studies included in the review (3 cross-sectional, 1 case-control and 3 cohorts) the NOS (Newcastle Otawa Scale) tool was used. Based on the Equator Network's recommendations for the quality and transparency of health research, it is recommended that the specific STROBE statement be used for each of these three types of studies. Therefore, we recommend that the authors to present the checklists corresponding to the STROBE statement.- The small number of studies included in the review (7 studies), some of them with poor methodological quality, would require evaluation of the evidence profile, such as, for example, the use of the GRADE PRO tool (free), through from which the evidence table could be prepared: method of presenting the quality of the available evidence. These judgments support the quality rating and the effects of management alternatives on the outcomes of interest. There are two formats (with repetition) available that serve different purposes and target different audiences: GRADE Evidence Profile and Summary Findings Table (SoFt).
- Assessment of risk of bias with REVMAN, a free tool from the Cochraine Collaboration, through which the individual risk of each of the studies must be detailed according to biases such as the absence of masking/concealment in allocation, absence of masking, incomplete patient count and outcomes, selective reporting of outcomes, other limitations. The authors are recommended to include the individual and joint evaluation of the risk of bias of the studies.
- A very important limitation exposed by the authors has been the impossibility of carrying out specific analyzes for the gender and age variables that could help to carry out a meta-regression. However, the authors considered performing meta-regression based on explanatory variables of an occupational nature (position, seniority, risk assessment of manual handling of loads or patients...)
Author Response
Point 1: For the evaluation of the quality of the studies included in the review (3 cross-sectional, 1 case-control and 3 cohorts) the NOS (Newcastle Ottawa Scale) tool was used. Based on the Equator Network's recommendations for the quality and transparency of health research, it is recommended that the specific STROBE statement be used for each of these three types of studies. Therefore, we recommend that the authors to present the checklists corresponding to the STROBE statement.
Response 1: We agree with the reviewer and we are familiar with the Equator Network's recommendations. However, for practicality reasons we prefer to work with NOS in the light of the few studies we were able to identify. Using three different quality assessment tools for seven studies seems to be confusing for us. Therefor we chose NOS because this instrument is well suited for assessing the quality of non-randomized studies, including cross-sectional (adapted), case-control and cohort studies.
Point 2: The small number of studies included in the review (7 studies), some of them with poor methodological quality, would require evaluation of the evidence profile, such as, for example, the use of the GRADE PRO tool (free), through from which the evidence table could be prepared: method of presenting the quality of the available evidence. These judgments support the quality rating and the effects of management alternatives on the outcomes of interest. There are two formats (with repetition) available that serve different purposes and target different audiences: GRADE Evidence Profile and Summary Findings Table (SoFt).
Response 2: We agree with the reviewer and considered using the GRADE tool ourselves. However, we came to the conclusion that the quality assessment of the studies + the assessment of the risk of bias are already meaningful. However, we came to the conclusion that the quality assessment of the studies + the assessment of the risk of bias are already meaningful. Using three different tools for seven studies seems to be confusing for us.
Point 3: Assessment of risk of bias with REVMAN, a free tool from the Cochraine Collaboration, through which the individual risk of each of the studies must be detailed according to biases such as the absence of masking/concealment in allocation, absence of masking, incomplete patient count and outcomes, selective reporting of outcomes, other limitations. The authors are recommended to include the individual and joint evaluation of the risk of bias of the studies.
Response 3: We have additionally performed the risk of bias assessment with REVMAN. We added the following to the results: …the risk of a selection and a performance bias was rated as high in all included studies, due to the fact that no randomisation based on the research question was possible. The allocation was based on the job title and could therefore not be concealed. The risk of detection bias could not be assessed in any of the studies because no sufficient information on the outcome assessment was reported. In two studies [48,50], an attrition bias was found to be high due to losses in follow-up surveys. A reporting bias was rated as low in all studies. A detailed description of the risk of bias assessment is provided in Supplementary file 4.
Point 4: A very important limitation exposed by the authors has been the impossibility of carrying out specific analyses for the gender and age variables that could help to carry out a meta-regression. However, the authors considered performing meta-regression based on explanatory variables of an occupational nature (position, seniority, risk assessment of manual handling of loads or patients...)
Response 4: We agree with the reviewer that it is a limitation of the review not to be able to perform more indebts meta-analysis. We believe this is clearly mentioned in the discussion of the limitations.
Reviewer 2 Report
Overall, to my knowledge and assessment, this is a good review paper.
as also noted by the authors, the analysis of these type of studies requires caution. The results of one type of hospital during a specific decade are unlikely generalisable to other hospital during other time period. The physical tasks of nurses in Taiwan and in Finland, for example, may be quite different. In some countries, the moving of patients are performed by porters rather than nurses.
What may be true thirty years ago may no longer be true nowadays.
If we accept ‘too often spine-burdening occupational activities’ can lead to LDD, then, I would think we already have the answer. Nurses who are too often engaged with moving/carrying the patients might have associated LDD, while nurses who are only occasionally or rarely engaged moving/carrying the patients may be fine with work-related LDD. Nurses should avoid these activities if possible and nurses should take preventative measures.
Pls note, the authors of initial studies might be keen to show positive results. How did they control the ‘outside work activities’, ? Personnel with sedentary jobs might have been engaged other physical activities after work. On the other hand, nurses may spend lots of their working hours standing and walking around, which itself are stressing. What would be the ideal control groups?
I feel some of the terms are not well defined, particularly disc aging vs, disc degeneration. I also feel disc bulging, protrusion, and herniation have not been precisely defined in the introduction. How do you define LDD on imaging (rather than physiological aging)?
Pls note the follow feature of low back pain (LBP) and LDD:
Most patients with acute LBP, with or without radiculopathy, have substantial improvements in pain and function in the first 4 weeks. Most cases of radiculopathy are self-limiting and symptoms resolve over the course of weeks to months.
Most cases of disc herniation reabsorb or regress by 8 weeks after symptom onset.
Imaging features of degenerative spine disease are common in asymptomatic individuals and increase with age. Disc height loss and disk bulge are moderately prevalent among young individuals, and their prevalence increases by approximately 1% per year.
Degenerative findings on MR imaging are not necessarily associated with the presence or the degree of LBP.
Size and type of disc herniation and location and presence of nerve root compression, which were important in terms of morphologic alteration, are not related to patient outcome.
Modic MT, et al. Radiology 2005;237:597-604. Brinjikji W, et al. AJNR Am J Neuroradiol 2015; 36(4):811-6. Wáng YXJ, et al, J Orthop Translat. 2018;15:21‐34.
Author Response
Point 1: as also noted by the authors, the analysis of these type of studies requires caution. The results of one type of hospital during a specific decade are unlikely generalisable to other hospital during other time period. The physical tasks of nurses in Taiwan and in Finland, for example, may be quite different. In some countries, the moving of patients are performed by porters rather than nurses.
What may be true thirty years ago may no longer be true nowadays.
Response 1: Thank you for the comment. That's an interesting point. However, there is one point on which we would like to disagree with you. While we share the view that moving of patients are indeed carried out by porters (see Savage et al.), the care of patients is usually carried out by nurses. Now we added the following to the discussion: “It may be difficult to compare the results of one hospital with other hospital types. There might be differences in the working methods of different countries or working structures may have changed over the decades, so that, for instance, nurses 30 years ago were much more exposed than they are nowadays.”
Point 2: If we accept ‘too often spine-burdening occupational activities’ can lead to LDD, then, I would think we already have the answer. Nurses who are too often engaged with moving/carrying the patients might have associated LDD, while nurses who are only occasionally or rarely engaged moving/carrying the patients may be fine with work-related LDD. Nurses should avoid these activities if possible and nurses should take preventative measures.
Response 2: Thank you for this conclusion. We think our review shows that preventive measures are important. We added this topic to the discussion: “HP, which is too often engaged in moving or carrying patients may have associated LDD, while nurses who are only occasionally or rarely engaged in patient handling may not be associated with LDD. HP should take technical or organizational preventive measures. Small or technical patient handling aids might relieve the lower back, however using these methods require considerably more time for nursing care than currently scheduled. Especially the preparation and postprocessing when using technical aids seems to be very time-consuming, in contrast to the actual implementation [55]. Another approach is to avoid or minimize these activities if possible, which could require the deployment of more staff.”
Point 3: Pls note, the authors of initial studies might be keen to show positive results. How did they control the ‘outside work activities’, ? Personnel with sedentary jobs might have been engaged other physical activities after work. On the other hand, nurses may spend lots of their working hours standing and walking around, which itself are stressing. What would be the ideal control groups?
Response 3: This is a very interesting point. We added this to the discussion: “Another limitation of the included studies is the confounder control. Nurses may spend much of their working time standing and walking around, which itself are stressing. Also, both exposed and non-exposed workers through patient handling may have been engaged in outside work activities. Staff with sedentary jobs may have engaged in other physical activities after work. We assume that LBP and LDD are caused by multiple factors. This renders the analysis of risk factors difficult and non-differential misclassification is a likely bias. In order to eliminate these confounding factors, both occupational and non-occupational spine straining activities would have to be considered for both groups.”
Point 4: Pls note the follow feature of low back pain (LBP) and LDD:
Most patients with acute LBP, with or without radiculopathy, have substantial improvements in pain and function in the first 4 weeks. Most cases of radiculopathy are self-limiting and symptoms resolve over the course of weeks to months.
Most cases of disc herniation reabsorb or regress by 8 weeks after symptom onset.
Imaging features of degenerative spine disease are common in asymptomatic individuals and increase with age. Disc height loss and disk bulge are moderately prevalent among young individuals, and their prevalence increases by approximately 1% per year.
Degenerative findings on MR imaging are not necessarily associated with the presence or the degree of LBP.
Size and type of disc herniation and location and presence of nerve root compression, which were important in terms of morphologic alteration, are not related to patient outcome.
Modic MT, et al. Radiology 2005;237:597-604. Brinjikji W, et al. AJNR Am J Neuroradiol 2015; 36(4):811-6. Wáng YXJ, et al, J Orthop Translat. 2018;15:21‐34.
Response 4: We are thanking for the proposed literature and we added this topic to the discussion: “Degenerative findings in MRI are not necessarily associated with the presence or severity of LBP. The size and type of herniated disc and the location and presence of nerve root compression, which were important in terms of morphological changes, are not related to the patient's outcome [59]. Most patients with acute LBP, with or without radiculopathy, show significant improvements in pain and function in the first four weeks. Most cases of radiculopathy are self-limiting, and symptoms subside over weeks to months. Most cases of herniated discs resorb or recede eight weeks after symptoms begin [60]. Radiologic alterations are not always clinically relevant. However typical clinical symptoms and disc alterations are associated. Therefore, it is noteworthy to study whether alterations of the discs occur more frequently or earlier in certain working groups.”
… There is ample evidence that LDD is more prevalent in the older population [52,53]. Compared to adults of working age, older adults are more likely to develop certain LBP pathologies, such as osteoporotic vertebral body fractures, tumors, spinal infections and lumbar spinal stenosis. [54].
Reviewer 3 Report
General Comments
The present manuscript aims to conduct a literature review to examine the prevalence or incidence of data related to work on LDD in health professionals. It is a relevant topic with a methodological approach conducted by a meta-analysis.
The authors conducted the study with good scientific writing and adequate methodological rigor for the proposal. Despite this, I suggest some minor observations in order to further improve the quality of the study.
INTRODUCTION
The Introduction is well described and clearly defines the problem to be researched. In addition, it justifies the relevance of the study by highlighting the unprecedented nature of the manuscript. However, I suggest that the authors reorganize the aspects related to the definition of the disease and then start to address the associated factors that contribute to the occurrence of the disease.
MATERIALS AND METHODS
This section is well described with the proper application of the main concepts. My only suggestion in this section would be just a better definition of the period in which the survey was conducted.
DISCUSSION
The discussion is written clearly and objectively. However, within the results that the authors reported that the subjects of all studies were generally composed of young in age. On the other hand, the literature is ample to state that LDD usually occurs mainly in older populations. In this sense, I think the authors could expand this issue further in their discussion. Another point that I think can be expanded refers to aspects related to work conditions and organizations as factors associated with the researched outcome.
Author Response
Point 1: INTRODUCTION
The Introduction is well described and clearly defines the problem to be researched. In addition, it justifies the relevance of the study by highlighting the unprecedented nature of the manuscript. However, I suggest that the authors reorganize the aspects related to the definition of the disease and then start to address the associated factors that contribute to the occurrence of the disease.
Response 1: Thank you for this important remark. Now we added that the diagnosis of LDD is an art. We added the following sentence to the introduction: LDD is not easy to diagnose, as radiologic alterations of the disc do not necessarily mean disease. They become a disease when they are associated with typical symptoms.
Point 2: MATERIALS AND METHODS
This section is well described with the proper application of the main concepts. My only suggestion in this section would be just a better definition of the period in which the survey was conducted.
Response 2: Thank you for this comment. Now we explain that we had no time restriction for the studies included: "A systematic literature search was conducted using the following databases from their beginning up to 2020/01/10. Therefore, no time frame other than the one introduced by the documentation systems was defined for the eligible studies".
Point 3: The discussion is written clearly and objectively. However, within the results that the authors reported that the subjects of all studies were generally composed of young in age. On the other hand, the literature is ample to state that LDD usually occurs mainly in older populations. In this sense, I think the authors could expand this issue further in their discussion. Another point that I think can be expanded refers to aspects related to work conditions and organizations as factors associated with the researched outcome.
Response 3: Thank you for this comment. We added this topic to the discussion: There is ample evidence that LDD is more prevalent in the older population [52,53]. Compared to adults of working age, older adults are more likely to develop certain LBP pathologies, such as osteoporotic vertebral body fractures, tumors, spinal infections and lumbar spinal stenosis. [54]. An earlier onset of illness in exposed workers is also an important observation that justifies preventive measures.
… HP, which is too often engaged in moving or carrying patients may have associated LDD, while nurses who are only occasionally or rarely engaged in patient handling may not be associated with LDD. HP should take technical or organizational preventive measures. Small or technical patient handling aids might relieve the lower back, however using these methods require considerably more time for nursing care than currently scheduled. Especially the preparation and postprocessing when using technical aids seems to be very time-consuming, in contrast to the actual implementation [55]. Another approach is to avoid or minimize these activities if possible, which could require the deployment of more staff.
Reviewer 4 Report
The article is very interesting and I recommend some revisions:
Line 27-28: I would remove this sentence
Line 165, 199, 210, 266, 298, 309: The sentence: (Error! Reference source not found, what is the meaning?
Line 300, 301, 369: References are missing, after writing the authors name.
Line 392, paediatrics is wrong spelled.
Line 442, generalise is wrong spelled.
The conclusion should be more direct and concise
Author Response
Point 1: Line 27-28: I would remove this sentence
Response 1: Yes, we agree. We deleted this sentence.
Point 2: Line 165, 199, 210, 266, 298, 309: The sentence: (Error! Reference source not found, what is the meaning?
Response 2: The problem resulted from missing links to tables and figures and is now fixed.
Point 3: Line 300, 301, 369: References are missing, after writing the authors name.
Response 3: Thank you, we have added the references.
Point 4: Line 392, paediatrics is wrong spelled.
Response 4: Thanks, we have corrected the spelling
Point 5: Line 442, generalise is wrong spelled.
Response 5: Thanks, we have corrected the spelling
Point 6: The conclusion should be more direct and concise
Response 6: We agree, the conclusions has now been amended by the additions of the discussion.
Round 2
Reviewer 1 Report
The authors have conveniently addressed the comments requested in the first review.
Therefore, I consider that the article has improved significantly, and I recommend its acceptance for publication.